# Relationship between Fasting Times and Emergence Delirium in Children Undergoing Magnetic Resonance Imaging under Sedation

**DOI:** 10.3390/medicina58121861

**Published:** 2022-12-16

**Authors:** Ayşe Neslihan Balkaya, Canan Yılmaz, Çağdaş Baytar, Asiye Demirel, Nermin Kılıçarslan, Filiz Ata, Nurcan Kat Kaçmaz, Mehmet Gamlı

**Affiliations:** 1Department of Anesthesiology and Reanimation, Bursa Yuksek Ihtisas Training and Research Hospital, Health Sciences University, Bursa 16310, Turkey; 2Department of Anesthesiology and Reanimation, Zonguldak Bülent Ecevit University Medicine Faculty, Zonguldak 67600, Turkey; 3Department of Radiology, Bursa Yuksek Ihtisas Training and Research Hospital, Health Sciences University, Bursa 16310, Turkey

**Keywords:** emergence delirium, pediatric, preoperative fasting, magnetic resonance imaging

## Abstract

*Background and Objectives*: This study aimed to determine whether there is a relationship between preoperative fasting time, fasting blood glucose (FBG), and postoperative emergence delirium (ED) in pediatric patients undergoing MRI under sedation. *Materials and Methods*: 110 pediatric patients were included in the study. Preoperative fasting (solid-fluid) time and FBG were recorded. The development of ED in the patients who underwent MRI under sedation was evaluated with the pediatric anesthesia emergence delirium (PAED) value for 30 min every 5 min in the recovery room. PAED score of ≥10 was grouped as having ED, and a PAED score of <10 as without ED at any time. The PAED scores were compared with other variables, ASA, age, weight, MRI examination time, and FBG level and fasting times. The risk factors affecting the occurrence of ED were examined. *Results*: Mean age was 3.94 ± 1.53 years, mean FBG was 106.97 ± 12.53 mg/dL, fasting time was 10.75 ± 2.61 h, solid food fasting time was 11.92 ± 2.33 h, and thirst time was 10.74 ± 2.58 h. FBG was never associated with PAED measurement at any time (*p* > 0.05). There was a weak positive correlation between the fasting time and the 0th, 5th, and 10th minute PAED score (r = 0.225; *p* = 0.018, r = 0.195; *p* = 0.041, r = 0.195; *p* = 0.041). There was a weak positive correlation between the solid food fasting time and the PAED score at the 0th, 5th, 10th, 15th, and 20th minutes (r = 0.382; *p* < 0.001, r = 0.357; *p* < 0.001, r = 0.345; *p* < 0.001, r = 0.360; *p* < 0.001, r = 0.240; *p* < 0.001). There was a weak positive correlation between thirst time and the PAED score at the 0th, 5th, and 10th minutes (r = 0.222; *p* = 0.020. r = 0.192; *p* = 0.045, r = 0.199; *p* = 0.037). The incidence of ED at any time was 34.5%. *Conclusions*: Prolonged fasting time, solid food fasting time and thirst time are risk factors for developing postoperative ED in children undergoing MRI under sedation.

## 1. Introduction

Magnetic resonance imaging (MRI) is a common pediatric imaging method with high soft tissue contrast resolution, providing imaging without ionizing radiation [1,2]. Long imaging times and imaging in confined spaces with loud noise make this method difficult for patients [3]. Sedation and general anesthesia are often used in MRI to achieve the best image quality, mainly to keep pediatric patients still [2,3].

Preoperative fasting is when liquid and/or solid food intake is not allowed for a defined time interval to ensure gastric emptying before the surgical procedure and reduce the incidence of gastric aspiration. The American Society of Anesthesiologists has a standard preoperative fasting guide for procedures requiring general anesthesia and sedation, and the term “preoperative” is considered synonymous with “pre-procedure” in this guide [4]. Despite the recommendations of the guidelines, prolongation of preoperative fasting can be seen frequently, especially in outpatient locations. In prolonged fasting, attention should be paid to fasting times, and prolonged fasting should be avoided, as hypoglycemia, dehydration, metabolic disturbances, irritability, fatigue, and drowsiness may develop in the pediatric population [5]. With the prolonged fasting time, there is typically a child who is irritable, annoyed, uncooperative, does not make eye contact, cannot be consoled by the parent, cries, and kicks [6]. Crying, irritability, and inability to be consoled in the preoperative period can facilitate the development of postoperative emergence delirium (ED) in the child [7,8]. Pediatric ED is a condition manifested by impaired awareness of the environment, disorientation of place, time, and person, hyperactive behaviors, and overreaction to external stimuli in the early period after anesthesia [6,9]. The prevalence of ED worldwide varies between 18% and 80%. Different descriptive criteria and scales are used in diagnosing ED [10,11]. In 2004, the pediatric anesthesia emergence delirium (PAED) scale, including cognitive evaluation components in addition to agitation behaviors, was developed, and the validity and reliability of this scale have been proven [10]. In a study of anesthesia-early delirium in children by Bong et al., a score of ≥10 on the PAED scale was shown to have the most significant sensitivity and specificity for diagnosing anesthesia-early delirium [12].

In addition, ED is a problem that hinders the recovery of the child and complicates the evaluation and management after anesthesia. Children’s aggressive and restless behaviors negatively affect the parents, causing restlessness and anxiety and lowering their satisfaction levels.

There are studies in the literature on preoperative fasting and postoperative ED or behavioral changes in children [7,8,9,10,11,12,13]. However, there was no study focusing on solid and fluid fasting times and their effectiveness in the development of postoperative ED in pediatric patients undergoing MRI under sedation. Our study hypothesized that the preoperative fasting time is a risk factor in the development of ED in children. Our study aimed to determine whether there is a relationship between preoperative fasting time and ED in pediatric patients undergoing MRI under sedation.

## 2. Materials and Methods

After approval by the Local Ethics Committee (Ref No 2011-KAEK-25 2021/12-07, ClinicalTrials.gov identifier: NCT05567718) and obtaining informed parental consent, children in the age group 2–6 years of American Society of Anesthesiologists (ASA) physical status I–III who underwent MRI under sedation were included in the study. Our prospective, observational study was conducted following the principles of the Declaration of Helsinki between January and June 2022 in Bursa Yuksek Ihtisas Research and Training Hospital. Those under the age of 2 or over the age of 6, those with ASA IV, those with cognitive or psychological disorders, those with central nervous system diseases, those with developmental delays, those with type 1 diabetes, those with a fasting blood glucose (FBG) ≤ 50 mg/dL, and those whose parents did not want to participate were excluded from the study. An anesthesiologist evaluated all patients the day before the MRI, giving information about the desired preoperative fasting times. It was explained that a fasting period of 6 h was required for preoperative solid food, formula food, and animal milk, and of 2 h for water. On the day of the MRI, the patients were taken to the premedication room accompanied by their parents. The time from the last oral intake to the start of the procedure was calculated and recorded in hours as the preoperative fasting time of the patients included in the study, without distinguishing between solid or liquid food. The time between the last oral clear liquid food intake and the start of the procedure was recorded in hours as the thirst time, and the time from the last oral solid food (solid food, formula, and animal-derived milk) intake to the beginning of the procedure was evaluated as the solid food fasting time and was recorded in hours. Solid food and fluid fasting time were recorded. Premedication was not administered to the patients, considering the possibility that the drugs to be used might affect the development of postoperative ED. In order not to affect preoperative fasting, patients were not given any liquid, either orally or intravenously (IV), that would provide energy and calories in the preoperative period. FBG was measured by a glucometer after IV vascular access was established. A 0.05–0.1 mg/kg dose of midazolam (Zolamid^®^, Defarma, Ankara, Turkey) as IV sedation and 1–3 mg/kg propofol IV (Propofol 1% Fresenius^®^, Fresenius Kabi, Bad Hamburg, Germany) were given to patients admitted to the MRI room. Oxygen saturation (SpO_2_) and heart rate (HR) were monitored using a pulse oximeter, and respiratory rate (RR) with a respiration band during the MRI scan. The patient was permitted to receive oxygen with a face mask throughout the procedure. If necessary, 0.5 mg/kg IV additional dose of propofol was administered. The patient was monitored with SpO_2_ and HR in the recovery room for 30 min following the end of the procedure. ED of all patients was evaluated by a blinded nurse working in the recovery room and trained in using the PAED scale. PAED scoring was first evaluated after the patient started awakening and was repeated every 5 min until the patient was discharged from the recovery room. A value greater than or equal to 10 at any time was considered ED. When a modified Aldrete score of >9 was obtained, children were observed for 30 min more and then transferred from the recovery room.

### Statistical Analysis

While calculating the sample size, the “large effect size” defined by Cohen was used during the calculations since the researcher did not have any predictions about the parameters to be used in the calculation or there was no reference study in the literature that could be used to obtain the parameters. The sample size required for the study was 98, with effect size = 0.60 for the Friedman test, providing an 83% power test and 95% confidence interval. This sample size also included the sample sizes required for other analysis methods to be used in the study. The relevant calculation was performed with the G *Power 3.1.9.2 package program.

Statistical analyses of the study were performed with R-studio Version 1.4.1717 (R Studio: Integrated Development for R, Boston, MA) software. Descriptive statistics of qualitative variables were given as frequency and percentage, and quantitative variables were given as mean, standard deviation, median, minimum and maximum values. The conformity of quantitative variables to normal distribution was examined using the Shapiro–Wilk test. The Mann–Whitney U test was used for group comparisons of independent variables, and Yates chi-square test was used for independent group comparisons of qualitative variables. Relationships between quantitative variables were evaluated with Spearman’s correlation coefficient. The variation of quantitative variables at different times was analyzed with the Friedman test. Logistic regression analysis was used to examine the risk factors affecting the occurrence of ED. The risk coefficients of the significant variables are given with 95% confidence intervals. *p* < 0.05 was considered statistically significant.

## 3. Results

One hundred fifteen patients were included in the study, and 110 completed the study (Figure 1). The demographic characteristics of the patients are given in Table 1. The mean age was 3.94 ± 1.53 years. Mean body weight (kg) was 17.01 ± 6.48, and FBG (mg/dL) was 106.97 ± 12.53. Twenty patients (18.2%) were using antiepileptic drugs. MRI duration was 18.98 ± 8.29 min, Fasting time was 10.75 ± 2.61 h, solid fasting time was 11.92 ± 2.33 h, and fluid fasting time was 10.74 ± 2.58 h.

No statistically significant relationship was found between FBG and fasting time and solid food fasting time (*p* = 0.186, *p* = 0.207). It was also observed that there was no relationship between thirst time and FBG (*p* = 0.188).

The incidence of postoperative ED was 34.5%. PAED scores and modified Aldrete scores are given in Table 2. The change in PAED values from the 0th to the 30th minute was statistically significant (*p* < 0.001). Only the change between the 20th and 25th minute and the change between the 25th and 30th minute were not significant (*p* > 0.05). The changes between all other times were statistically significant (*p* < 0.05).

A PAED score of ≥10 was grouped as having ED, and a PAED score of <10 as without ED, and when compared with other variables, ASA, age, weight, MRI examination time, and FBG level were found to be similar in children with or without ED (*p* = 0.716, *p* = 0.933, *p* = 0.327, *p* = 0.466, *p* = 0.391, respectively). It was observed that the fasting time, solid food fasting time, and fluid fasting time were longer in patients with ED than without ED (*p* = 0.027, *p* <0.001, *p* = 0.030, respectively) (Table 3). In the regression analysis, the effect of solid food fasting time on the presence of ED was significant (*p* = 0.007), while the effect of fluid fasting time was not significant in the model (*p* = 0.760). A 1-unit increase in the solid fasting time variable increased the risk of ED presence 1.574 fold (Table 4).

A weak positive correlation was found between the fasting time and the PAED scores at the 0th, 5th, and 10th minutes (r = 0.225 *p* = 0.018, r = 0.195 *p* = 0.041, r = 0.202 *p* = 0.034, respectively). A weak positive correlation was found between the solid food fasting time and the PAED scores at the 0th, 5th, 10th, 15th, and 20th minutes. While there was a weak positive correlation between thirst time and PAED scores at the 0th, 5th, and 10th minutes, no correlation was found between FBG and PAED measurements. Correlation values of fasting and thirst times are given in Table 5 in detail.

## 4. Discussion

Postoperative ED and agitation are common in pediatric patients, and many studies exist about their causes. General anesthesia, major surgery, resection of large areas, prolonged duration of anesthesia, postoperative pain, and using opioids are the most common predisposing factors in the development of postoperative ED [6,14,15]. Preoperative thirst and fasting can also cause behavioral changes such as anger, agitation, and delirium in pediatric patients. Few studies have evaluated the relationship between preoperative fasting time and postoperative behavior change [7,16]. In the study by Xara et al. in 2013, it was reported that prolonged preoperative fasting was a predisposing factor for postoperative agitation and delirium [17]. In the study by Chauvin et al., it was reported that prolonged thirst may cause behavioral changes in children in the postoperative period [13]. In the study by Khanna et al., it was reported that postoperative ED increased with prolonged fasting [7]. In studies on preoperative fasting times and ED development, the distinction between solid and fluid food fasting times and the difference between these times in the development of ED have not been clearly emphasized. In our study, we focused on solid food and fluid fasting times and their effects on the development of postoperative ED.

In the current study with the hypothesis that preoperative fasting time and fasting blood glucose levels are facilitating factors in the development of postoperative ED in children, pediatric patients who underwent MRI under sedation were selected to avoid other predisposing factors in the development of postoperative ED. Studies of ED in patients undergoing MRI are particularly useful because they eliminate pain as a confounding variable and allow for a more controlled investigation of ED. The absence of pain and general anesthesia, a short process, the short duration of anesthesia, and the absence of surgical stress allow the examination of the relationship between ED and fasting in pediatric patients in the MRI procedure. In the literature, there are studies of preoperative fasting and postoperative ED in pediatric patients undergoing various surgeries, but the effect of the two factors on each other has not been evaluated [7,16]. To the best of our knowledge, the current study is the first to evaluate the relationship between preoperative fasting and postoperative ED in pediatric patients undergoing MRI.

The effect of preoperative fasting on FBG concentration in pediatric patients is controversial. In the study by Thomas et al., hypoglycemia was observed after long-term preoperative fasting, while in other studies, there was no relationship between preoperative fasting time and blood sugar [7,18,19]. Hypoglycemia is provoked by fasting, though it is not an inevitable consequence of withholding food. The body conserves glucose by decreasing the concentration of circulating insulin and increasing the concentration of counter-regulatory hormones–growth hormones, glucagon, cortisol, and adrenaline [19]. In our study, it was hypothesized that preoperative fasting may affect fasting blood glucose in children, and this situation may also affect the development of postoperative ED. However no relationship was found between preoperative fasting time and FBG, and the lowest FBG was 76 mg/dL in the patients participating in the study. In our pediatric patient group, no patient had hypoglycemia after preoperative fasting. In our study, no relationship was found between fasting blood glucose levels and delirium. It was observed that PAED scores increased in the first 10 min of measurements in the recovery room with the prolongation of the preoperative food and fluid fasting times and that PAED scores could be higher up to the first 20 min in those with prolonged solid food fasting. It was determined that the prolongation of the preoperative fasting period is a facilitating factor for the development of postoperative ED in children. It was found that the prolongation of the solid food fasting period may cause the prolongation of the postoperative ED period. While a positive correlation was found between the preoperative fasting and thirst times and the PAED scores in the first 10 min of measurements in the recovery room, a significant positive correlation was found between the solid food fasting time and the PAED scores in the first 20 min of measurements. It was observed that the prolongation of the preoperative fasting period was a facilitating factor for the development of postoperative ED in children.

Agitation in the preoperative period, prolonged fasting, hypovolemia, hypoglycemia, and dissatisfaction may increase the development of agitation and delirium in the postoperative period in children [19]. Intolerance to hunger, irritability, and crying due to hunger and thirst are common conditions in children. Especially, the longer the solid food fasting period, the more intense the sensation of hunger that the child feels. Children may be restless in the preoperative period with the feeling of hunger and thirst, and this may be reflected in the postoperative period. We consider that the relationship between preoperative fasting, especially prolonged solid food fasting, and the development of postoperative ED, is related to this situation.

We found a 34.5% incidence of postoperative ED, considering a PAED score of more than 10. There are different rates of the incidence of ED in the literature. Costi et al. reported a 29% incidence of postoperative ED in children undergoing MRI scans [20]. Khanna et al. reported a postoperative ED incidence of 24%, and Baek et al. of 33.3% [7,21]. The lack of routine premedication to outpatients in our hospital may have affected the incidence of ED and caused it to be higher than the levels reported in the literature.

The incidence of pulmonary aspiration of gastric contents, which is an important complication of anesthesia, is 0.07–0.1% [22]. A longer preoperative fasting time is recommended for solid foods than for liquids in elective cases to minimize the risk of pulmonary aspiration, a major complication [22,23]. In the guidelines of the anesthesia societies, the “6-4-2 rule” (6 h for solid food, formula food, and animal-derived milk, 4 h for breast milk, and 2 h for clear liquids) is recommended for preoperative fasting times [4,24]. Due to the risk of pulmonary aspiration, the anxiety of the physicians and the disruptions in the routine functioning of the operating room (such as delay and postponement) can cause prolonged preoperative fasting times in patients without separating solid and liquid food. In studies evaluating preoperative fasting situations, it was observed that both adult and pediatric patients have prolonged fasting and thirst times [7,25,26]. In our prospective observational study, similarly to the literature, it was found that there were disruptions in MRI processing (such as prolonged imaging times and changes in imaging sequences), preoperative intake times of both liquid and solid food were longer than the times recommended in the guidelines, and fasting and thirst times were prolonged.

There are several limitations of our study. The preoperative anxiety level of the child or the parent may cause the development of restlessness in the child in the postoperative period. Children are easily affected by the emotional states of their parents. In this study, the preoperative anxiety levels of the child and the parent were not examined. Our study did not evaluate the child’s preoperative anxiety level, the existence of a relationship between anxiety and fasting time, or the effect of this situation on the development of postoperative ED.

Prolonged fasting time, solid food fasting time and thirst time are risk factors for developing postoperative ED in children undergoing MRI under sedation.

## Figures and Tables

**Figure 1 medicina-58-01861-f001:**
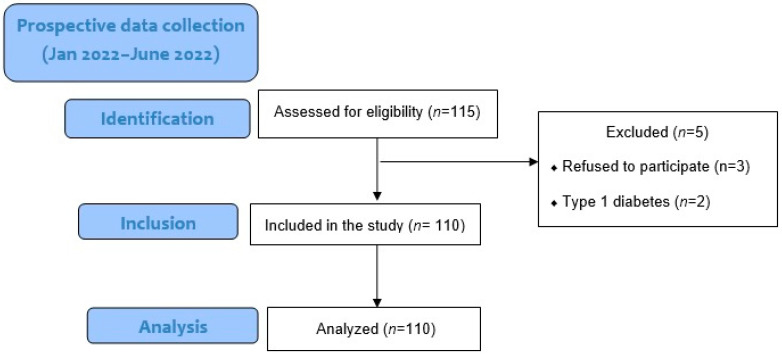
The study flow chart in line with the STROBE.

**Table 1 medicina-58-01861-t001:** Demographic and clinical data.

		*n* (%)
ASA	1	46 (41.8%)
	2	62 (56.4%)
	3	2 (1.8%)
Sex	Female	45 (40.9%)
	Male	65 (59.1%)
Diagnosis	Epilepsy	30 (27.3%)
	Puberty precocious	3 (2.7%)
	Nystagmus	5 (4.5%)
	Scoliosis	3 (2.7%)
	Neuromotor delay	21 (19.1%)
	Spina bifida	8 (7.3%)
	Autism	12 (10.9%)
	Intracranial tumor	9 (8.2%)
	Hydrocephalus	12 (10.9%)
	Headache	3 (2.7%)
	Strabismus	4 (3.6%)
MRI examination	Cranial	82 (74.5%)
	Cranial + cervical + thoracal vertebra	4 (3.6%)
	Lumbar vertebrae	7 (6.4%)
	Cervical + thoracal + lumbar vertebrae	5 (4.5%)
	Cranial + cervical + thoracal + lumbar vertebrae	1 (0.9%)
	Cranial + lumbar vertebrae	4 (3.6%)
	orbital + cranial	7 (6.4%)

ASA: American Society of Anesthesiologists physical status.

**Table 2 medicina-58-01861-t002:** Pediatric Anesthesia Emergence Delirium and Modified Aldrete Scores.

Minutes		Median *
0	PAED	8 (2–16)
	MAS	8 (0–10)
5	PAED	7 (1–15)
	MAS	9 (6–10)
10	PAED	6 (0–15)
	MAS	9 (7–10)
15	PAED	4 (0–15)
	MAS	10 (7–10)
20	PAED	2 (0–10)
	MAS	10 (7–10)
25	PAED	0 (0–8)
	MAS	10 (7–10)
30	PAED	0 (0–8)
	MAS	10 (9–10)

MAS: modified Aldrete score, PAED: pediatric anesthesia emergence delirium, *: (minimum–maximum).

**Table 3 medicina-58-01861-t003:** Presence of emergence delirium according to the patient’s characteristics.

		ED (−) (*n* = 72)	ED (+) (*n* = 38)	*p*
ASA		1.61 ± 0.52 2 (1–3)	1.58 ± 0.55 2 (1–3)	0.716 ^#^
Age		3.94 ± 1.49 4 (2–6)	3.95 ± 1.64 3.5 (2–6)	0.933 ^#^
Weight		16.53 ± 6.22 15 (8–39)	17.95 ± 6.95 15.5 (8–33)	0.327 ^#^
Duration of MRI examination		18.69 ± 8.02 15 (10–45)	19.53 ± 8.86 20 (10–50)	0.466 ^#^
FBG level		106.38 ± 13.03 106 (78–140)	108.11 ± 11.60 107.5 (76–131)	0.391 ^#^
Fasting time		10.38 ± 2.51 10.50 (6–15)	11.47 ± 2.69 11 (6–16)	**0.027** ^#^
Solid fasting time		11.32 ± 2.15 12 (6–16)	13.05 ± 2.25 13.5 (8–18)	**<0.001** ^#^
Fluid fasting time		10.38 ± 2.51 10.5 (6–15)	11.42 ± 2.62 12 (6–15)	**0.030** ^#^
Gender	Female Male	31 (43.1%) 41 (56.9%)	14 (36.8%) 24 (63.2%)	0.670 ^##^
Drug use	+ (Antiepileptic) −	15 (20.8%) 57 (79.2%)	5 (13.2%) 33 (86.8%)	0.464 ^##^
Additional disease	+ −	14 (19.4%) 58 (80.6%)	5 (13.2%) 33 (86.8%)	0.573 ^##^

^#^ Mann–Whitney U test, ^##^ Yates chi-square test, ED: emergence delirium, ASA: American Society of Anesthesiologists, MRI: magnetic resonance imaging, FBG: fasting blood glucose.

**Table 4 medicina-58-01861-t004:** Factors affecting the presence of emergence delirium.

	B	SE	Wald	df	*p*	Exp (B)	For Exp (B) 95% CI
Solid fasting time	0.454	0.168	7.320	1	**0.007** ^#^	1.574	1.133–2.187
Fluid fasting time	−0.044	0.145	0.093	1	0.760 ^#^	0.957	0.719–1.272
Constant	6.047	2.852	4.495	1	**0.034** ^#^	422.912	

^#^ Logistic Regression Analysis SE: standard error, CI: confidence interval.

**Table 5 medicina-58-01861-t005:** Correlation between pediatric anesthesia emergence delirium scores, fasting blood glucose, and fluid and solid food fasting durations at different time points.

	PAED0	PAED5	PAED10	PAED15	PAED20	PAED25	PAED30
Fasting time	r = 0.225 ^#^ *p* = 0.018 *	r = 0.195 ^#^ *p* = 0.041 *	r = 0.202 ^#^ *p* = 0.034 *	r = 0.185 ^#^ *p* = 0.053	r = 0.079 ^#^ *p* = 0.413	r = 0.058 ^#^ *p* = 0.550	r = 0.093 ^#^ *p* = 0.333
Solid fasting time	r = 0.382 ^#^ *p*< 0.001 **	r = 0.357 ^#^ *p*< 0.001 **	r = 0.345 ^#^ *p*< 0.001 **	r = 0.360 ^#^ *p*< 0.001 **	r = 0.240 ^#^ *p*< 0.001 **	r = 0.169 ^#^ *p* = 0.078	r = 0.156 ^#^ *p* = 0.103
Fluid fasting time	r = 0.222 ^#^ *p* = 0.020 *	r = 0.192 ^#^ *p* = 0.045 *	r = 0.199 ^#^ *p* = 0.037 *	r = 0.181 ^#^ *p* = 0.058	r = 0.074 ^#^ *p* = 0.442	r = 0.051 ^#^ *p* = 0.595	r = 0.086 ^#^ *p* = 0.369
Fasting blood glucose	r = 0.142 ^#^ *p* = 0.138	r = 0.110 ^#^ *p* = 0.253	r = 0.120 ^#^ *p* = 0.214	r = 0.093 ^#^ *p* = 0.332	r = 0.157 ^#^ *p* = 0.102	r = 0.15 ^#^ *p* = 0.117	r = 0.100 ^#^ *p* = 0.301

^#^ Spearman correlation test, * *p* value < 0.05, ** *p* value< 0.001, r: correlation ratio. PAED: pediatric anesthesia emergence delirium.

## Data Availability

The data presented in this study are available on request from the corresponding author.

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
