# Peer review of "Relationship between Fasting Times and Emergence Delirium in Children Undergoing Magnetic Resonance Imaging under Sedation"

_medicina, 2022, doi:10.3390/medicina58121861_

Round 1
Reviewer 1 Report
1. General comments:
a. Please insert full stop after the end of sentence but before the superscripted reference numbers (….cries and kicks.(6)) throughout the document.
b. Check the text for suggestions.
2. Title: This should read as “Relationship between fasting times and emergency delirium in children undergoing magnetic resonance imaging under sedation”
3. Abstract: The text in the abstract uses at places present and others past tense. Please use same tense throughout.
4. Introduction: The introduction should mention if a similar hypothesis has been investigated by other authors too and what was their conclusion. If this hypothesis has not been tested before this should be mentioned.
5. Material and methods:
a. Why was no distinction was maintained between solid and liquid fasting times?
b. What was the reason for not administering the premedication and what relationship is there between premedication and admission of the child.
6. Results: Adequately summarized
7. Discussion:
a. The presence of prolonged fasting especially solid food fasting led to increase delirium but this had no correlation with low blood glucose values. What do the authors think is the reason of this finding? What is the reason attributed to this finding? As rightly pointed out the assessment of preoperative anxiety in parents and the child needs to be done in such studies.
b. Other studies have also presented similar results in children undergoing MRI. What new is being studies by the authors in their study?
8. References: Correctly cited.

Author Response
First of all, thank you for the comments.
- General comments:
- Please insert full stop after the end of sentence but before the superscripted reference numbers (….cries and kicks.(6)) throughout the document.
---- All references cited in the article have been corrected.
- Check the text for suggestions.
---- We checked the text and all your suggestions are applied.
- Title:This should read as “Relationship between fasting times and emergency delirium in children undergoing magnetic resonance imaging under sedation”
---- The title has been edited as you suggested.
- Abstract:The text in the abstract uses at places present and others past tense. Please use same tense throughout.
---- Necessary corrections have been made.
- Introduction:The introduction should mention if a similar hypothesis has been investigated by other authors too and what was their conclusion. If this hypothesis has not been tested before this should be mentioned.
---- Added to the introduction as ‘’ There are studies in the literature on preoperative fasting and postoperative ED or behavioral changes in children [7-13]. However, there was no study focusing on solid and fluid fasting times and their effectiveness in the development of postoperative ED in pediatric patients undergoing MRI under sedation.’’
- Material and methods:
- Why was no distinction was maintained between solid and liquid fasting times?
---- Added to the Material Method as ’’ It was explained that a fasting period of 6 hours was required for preoperative solid food, formula food and animal milk, and 2 hours for water.’’
- What was the reason for not administering the premedication and what relationship is there between premedication and admission of the child.
---- Added to the Material Method as ‘’Solid food and fluid fasting time were recorded. Premedication was not administered to the patients, considering the possibility that the drugs to be used may affect the development of postoperative ED. ‘’
- Results: Adequately summarized
---- Necessary arrangments have been made.
- Discussion:
- The presence of prolonged fasting especially solid food fasting led to increase delirium but this had no correlation with low blood glucose values. What do the authors think is the reason of this finding? What is the reason attributed to this finding? As rightly pointed out the assessment of preoperative anxiety in parents and the child needs to be done in such studies.
---- Added to the discussion section as ‘’ In the literature, many factors are shown as risk factors for the development of postoperative delirium. Preoperative fasting is also considered as one of these factors. While the study was being planned, it was thought that long-term solid food fasting could lower fasting blood sugar in children, and this situation might affect the development of postoperative delirium. But the study results showed that the duration of solid food fasting did not affect fasting blood sugar in children. Hypoglycemia was not observed in study patients, including children with prolonged solid food fasting, and no correlation was found between fasting blood glucose level and delirium. Intolerance of hunger, irritability and crying due to hunger are common conditions in children. Especially as the solid food fasting period gets longer, the feeling of hunger that the child feels becomes more intense and gives uneasiness. We think that the relationship between fasting, especially solid food fasting, and the development of postoperative delirium depends on this.’’
- Other studies have also presented similar results in children undergoing MRI. What new is being studies by the authors in their study?
---- Added to the discussion as ‘’ In studies on preoperative fasting times and ED development, the distinction between solid and fluid food fasting times and the difference between these times in the development of ED have not been clearly emphasized. In our study, we focused on solid food and fluid fasting times and their effectiveness in the development of postoperative ED.’’
- References: Correctly cited.
---- Necessary arrangments have been made.
Reviewer 2 Report
Dear Authors,
I would like to thank you for your contribution. The manuscript “Relationship Between Fasting Time and Emergency Delirium in Pediatric Sedation Undergoing Magnetic Resonance Imaging” deals with a clinically relevant problem. Nevertheless, there are some major aspects that lead me to recommend an overall revision of the manuscript (of the statistical methodology in particular). I hope that my comments are useful and can help to improve your manuscript, as this topic is of high relevance.
Majors:
1. It is the goal of your study to investigate the “relationship between preoperative fasting time and ED.” Unfortunately, the statistical methodology you use is not appropriate to achieve this goal. You only performed correlation analyses between the parameters of fasting and the PAED scores. To investigate the relationship between parameters of fasting with ED (which is defined as a PAED score ≥ 10 points), you should have considered the cuf-off of the PAED score in your analyses. The most common statistical methodology used in this field is simple group comparison (between patients with and without ED) to describe patient characteristics and descriptive analysis of fasting parameters, followed by logistic regression analysis. Consequently, I strongly recommend a revision of your statistical methodology.
2. “While PAED scores were found to be high in the first 10-minute measurements in children with prolonged fasting and thirst time, PAED scores were higher in the first 20-minute measurements in those with a pro-longed solid food fasting time.” - from the statistical methodology you have used and presented, this conclusion cannot be drawn.
3. “It was determined that the longer the solid food fasting time, the longer the postoperative ED period.” - from the statistical methodology you have used and presented, this conclusion cannot be drawn.
4. From my point of view you should address the generalizability of your results in the discussion. While you have worded it very nicely: “In our study, pediatric patients who underwent MRI under sedation were selected in order to exclude conditions affecting the development of postoperative ED, such as general anesthesia, major surgery, large area resection, prolonged anesthesia time, presence of postoperative pain, and opioid use to examine the relationship between preoperative fasting time, last feeding time with solid and liquid food, FBG and postoperative ED.” - your argumentation is not very convincing. You should discuss to what extent your results from a cohort with an intervention that lasted only about 19 minutes and received only sedation is transferable to real-life situations in which ED is highly relevant. In the discussion, please also address in what way the patients' underlying diseases influenced the results. Patients who require MRI examinations are a very selective patient group. This should be taken into account at least in a simple group comparison and, if necessary, also in the regression analysis.
5. Please follow the general guidelines for presenting the results of a cohort study. Please, apply to the STROBE checklist (include a FlowChart, state the time period of the study conduction, description of all variables in the method section)
Minors:
Abstract:
1. Please include a description of the statistical methods in the abstract.
2. Please state units of fasting times
3. Please correct the mean age. This must be a typo.
Methods:
1. please specify more in detail “An anesthesiologist […] giving information about preoperative fasting times”
Results:
1. Please state incidence of postoperative ED in the results section
2. Please correct the mean age. This must be a typo.
3. According to Table 1: how can it be that there were patients with an ASA PS of 3 when patients with ASA PS of 3 and 4 were excluded?
4. According to Table 1: Please replace gender with sex (I assume you have examined biological sex).
5. Please provide more content on the “Drug Use” and “Comorbidities variables”. As it is presented now, it is not possible to get an impression of the cohort.
Author Response
First of all, thank you for the comments.
Majors:
- It is the goal of your study to investigate the “relationship between preoperative fasting time and ED.” Unfortunately, the statistical methodology you use is not appropriate to achieve this goal. You only performed correlation analyses between the parameters of fasting and the PAED scores. To investigate the relationship between parameters of fasting with ED (which is defined as a PAED score ≥ 10 points), you should have considered the cuf-off of the PAED score in your analyses. The most common statistical methodology used in this field is simple group comparison (between patients with and without ED) to describe patient characteristics and descriptive analysis of fasting parameters, followed by logistic regression analysis. Consequently, I strongly recommend a revision of your statistical methodology.
---- Statistical analysis revised and the new results added to the results section.
- “While PAED scores were found to be high in the first 10-minute measurements in children with prolonged fasting and thirst time, PAED scores were higher in the first 20-minute measurements in those with a pro-longed solid food fasting time.” - from the statistical methodology you have used and presented, this conclusion cannot be drawn.
----- The statistical methodology we used has been corrected and added to the statistical analysis section.
- “It was determined that the longer the solid food fasting time, the longer the postoperative ED period.” - from the statistical methodology you have used and presented, this conclusion cannot be drawn.
----- The statistical methodology we used has been corrected and added to the statistical analysis section.
- From my point of view you should address the generalizability of your results in the discussion. While you have worded it very nicely: “In our study, pediatric patients who underwent MRI under sedation were selected in order to exclude conditions affecting the development of postoperative ED, such as general anesthesia, major surgery, large area resection, prolonged anesthesia time, presence of postoperative pain, and opioid use to examine the relationship between preoperative fasting time, last feeding time with solid and liquid food, FBG and postoperative ED.” - your argumentation is not very convincing. You should discuss to what extent your results from a cohort with an intervention that lasted only about 19 minutes and received only sedation is transferable to real-life situations in which ED is highly relevant. In the discussion, please also address in what way the patients' underlying diseases influenced the results. Patients who require MRI examinations are a very selective patient group. This should be taken into account at least in a simple group comparison and, if necessary, also in the regression analysis.
----- The statistical methodology we used has been corrected and added to the statistical analysis section.
- Please follow the general guidelines for presenting the results of a cohort study. Please, apply to the STROBE checklist (include a FlowChart, state the time period of the study conduction, description of all variables in the method section)
---- ---- Necessary arrangments have been made.
Minors:
Abstract:
- Please include a description of the statistical methods in the abstract.
- Please state units of fasting times.---- stated.
- Please correct the mean age. This must be a typo.--- Corrected.
Methods:
- please specify more in detail “An anesthesiologist […] giving information about preoperative fasting times”.. ----- fasting times were given clearly.
Results:
- Please state incidence of postoperative ED in the results section. ---- Added to ther esult section.
- Please correct the mean age. This must be a typo.---- Corrected.
- According to Table 1: how can it be that there were patients with an ASA PS of 3 when patients with ASA PS of 3 and 4 were excluded?---- corrected
- According to Table 1: Please replace gender with sex (I assume you have examined biological sex). ---- corrected
- Please provide more content on the “Drug Use” and “Comorbidities variables”. As it is presented now, it is not possible to get an impression of the cohort. ---- excluded from the table because it did not contribute to the cohort
Round 2
Reviewer 2 Report
Dear Authors, thank you for making such extensive revisions. While some points have been significantly improved, others remain in need of revision. I hope that my comments are useful and can help to further improve your manuscript.
Regarding my previous comments:
Majors:
- It is the goal of your study to investigate the “relationship between preoperative fasting time and ED.” Unfortunately, the statistical methodology you use is not appropriate to achieve this goal. You only performed correlation analyses between the parameters of fasting and the PAED scores. To investigate the relationship between parameters of fasting with ED (which is defined as a PAED score ≥ 10 points), you should have considered the cuf-off of the PAED score in your analyses. The most common statistical methodology used in this field is simple group comparison (between patients with and without ED) to describe patient characteristics and descriptive analysis of fasting parameters, followed by logistic regression analysis. Consequently, I strongly recommend a revision of your statistical methodology.
---- Statistical analysis revised and the new results added to the results section.
è I think that your analyses regarding simple group differences contributes to a better understanding of the connections. However, the regression analysis does not seem to be sufficient. On what basis did you decide to include only solid fasting time and fluid fasting time in the model? A more detailed description of the model and its rationale would be desirable.
- “While PAED scores were found to be high in the first 10-minute measurements in children with prolonged fasting and thirst time, PAED scores were higher in the first 20-minute measurements in those with a pro-longed solid food fasting time.” - from the statistical methodology you have used and presented, this conclusion cannot be drawn.
----- The statistical methodology we used has been corrected and added to the statistical analysis section.
è What statistical adjustments have you made exactly? From my point of view, this statement is still not supported by the analyses you have performed. From my point of view, you just did correlation analyses between individual Fasting parameters and PAED scores at different time points. It cannot be concluded from these analyses that PAED scores were higher in those with a pro-longed solid food fasting time. Furthermore, you added the following sentences to the discussion, which are also not supported by your calculations: “It was observed that PAED scores increased in the first 10 minutes of measurements in the recovery room with the prolongation of the preoperative food and fluid fasting times and that PAED scores could be higher up to the first 20 minutes in those with prolonged solid food Fasting” and “It was found that the prolongation of the solid food fasting period may cause the prolongation of the post operative ED period”. I strongly recommend taking statistical advice.
- “It was determined that the longer the solid food fasting time, the longer the postoperative ED period.” - from the statistical methodology you have used and presented, this conclusion cannot be drawn.
----- The statistical methodology we used has been corrected and added to the statistical analysis section.
è The same here. See above.
- From my point of view you should address the generalizability of your results in the discussion. While you have worded it very nicely: “In our study, pediatric patients who underwent MRI under sedation were selected in order to exclude conditions affecting the development of postoperative ED, such as general anesthesia, major surgery, large area resection, prolonged anesthesia time, presence of postoperative pain, and opioid use to examine the relationship between preoperative fasting time, last feeding time with solid and liquid food, FBG and postoperative ED.” - your argumentation is not very convincing. You should discuss to what extent your results from a cohort with an intervention that lasted only about 19 minutes and received only sedation is transferable to real-life situations in which ED is highly relevant. In the discussion, please also address in what way the patients' underlying diseases influenced the results. Patients who require MRI examinations are a very selective patient group. This should be taken into account at least in a simple group comparison and, if necessary, also in the regression analysis.
----- The statistical methodology we used has been corrected and added to the statistical analysis section.
è That's okay from my point of view.
- Please follow the general guidelines for presenting the results of a cohort study. Please, apply to the STROBE checklist (include a FlowChart, state the time period of the study conduction, description of all variables in the method section)
---- ---- Necessary arrangments have been made.
è Unfortunately, a description of all collected items is still missing in the method section.
Minors:
Abstract:
- Please include a description of the statistical methods in the abstract.
è This did not happen. Nor was the abstract as a whole revised after the statistics were adjusted. This is not satisfactory.
- Please state units of fasting times.---- stated.
- Please correct the mean age. This must be a typo.--- Corrected.
Methods:
- please specify more in detail “An anesthesiologist […] giving information about preoperative fasting times”.. ----- fasting times were given clearly.
Results:
- Please state incidence of postoperative ED in the results section. ---- Added to ther esult section.
- Please correct the mean age. This must be a typo.---- Corrected.
- According to Table 1: how can it be that there were patients with an ASA PS of 3 when patients with ASA PS of 3 and 4 were excluded?---- corrected
- According to Table 1: Please replace gender with sex (I assume you have examined biological sex). ---- corrected
è Unfortunately, you continue to use "gender" in the manuscript
- Please provide more content on the “Drug Use” and “Comorbidities variables”. As it is presented now, it is not possible to get an impression of the cohort. ---- excluded from the table because it did not contribute to the cohort
Further Minors:
1. Please also provide number and percentages of ED (+) and ED (-) in Table 3.
2. Please do not use the * to annotate Mann Whitney U test. The * is very often equated to the significance level in publications.
3. You stated “In the literature, there are studies of preoperative fasting and postoperative ED in pediatric patients undergoing MRI, but the effect of the two factors on each other has not been evaluated.” Please provide adequate references.
4. Please stay concise with the term “ED”. In some places you use the term “postoperative delirium”.
Author Response
First of all, thank you for the comments.
We reply all comments as follow.
Majors:
- It is the goal of your study to investigate the “relationship between preoperative fasting time and ED.” Unfortunately, the statistical methodology you use is not appropriate to achieve this goal. You only performed correlation analyses between the parameters of fasting and the PAED scores. To investigate the relationship between parameters of fasting with ED (which is defined as a PAED score ≥ 10 points), you should have considered the cuf-off of the PAED score in your analyses. The most common statistical methodology used in this field is simple group comparison (between patients with and without ED) to describe patient characteristics and descriptive analysis of fasting parameters, followed by logistic regression analysis. Consequently, I strongly recommend a revision of your statistical methodology.
---- Statistical analysis revised and the new results added to the results section.
è I think that your analyses regarding simple group differences contributes to a better understanding of the connections. However, the regression analysis does not seem to be sufficient. On what basis did you decide to include only solid fasting time and fluid fasting time in the model? A more detailed description of the model and its rationale would be desirable.
----- ¥ According to the first revision request, the variables in the study were compared according to these groups by grouping according to the PAED score, and a regression model was established and interpreted with the variables that were significant. Accordingly, those with a PAED score of 10 and above were grouped as having ED, and those below 10 were grouped as no ED and compared with other variables in the study. In addition, a regression model was established with the variables found to be significant and the results are given in Table 3. Solid and liquid fasting times were statistically significant. Accordingly, the effect of solid and liquid fasting times on the presence of ED was evaluated with the logistic regression model, and the explanatory power of the model was low. Although there was no significant difference between those with and without ED when evaluated alone, when the variables of age, sex, additional disease, duration of MRI examination, FBG levels were added to the model, it was observed that these variables had no effect on the model and were excluded from the model. The regression model showing the effect of solid and liquid fasting times on the presence of ED is given in Table-4.
- “While PAED scores were found to be high in the first 10-minute measurements in children with prolonged fasting and thirst time, PAED scores were higher in the first 20-minute measurements in those with a pro-longed solid food fasting time.” - from the statistical methodology you have used and presented, this conclusion cannot be drawn.
----- The statistical methodology we used has been corrected and added to the statistical analysis section.
è What statistical adjustments have you made exactly? From my point of view, this statement is still not supported by the analyses you have performed. From my point of view, you just did correlation analyses between individual Fasting parameters and PAED scores at different time points. It cannot be concluded from these analyses that PAED scores were higher in those with a pro-longed solid food fasting time. Furthermore, you added the following sentences to the discussion, which are also not supported by your calculations: “It was observed that PAED scores increased in the first 10 minutes of measurements in the recovery room with the prolongation of the preoperative food and fluid fasting times and that PAED scores could be higher up to the first 20 minutes in those with prolonged solid food Fasting” and “It was found that the prolongation of the solid food fasting period may cause the prolongation of the post operative ED period”. I strongly recommend taking statistical advice.
----- ¥ Statistical support was received after your suggestions.
¥ ¥ PaED scores in the recovery room and the timing relationship were reinterpreted. The statement added to the manuscript as ''While a positive correlation was found between the preoperative fasting and thirst times and the PAED scores in the first 10-min measurements in the recovery room, a significant positive correlation was found between the solid food fasting time and the PAED scores in the first 20-min measurements. It was observed that the prolongation of the preoperative fasting period was a facilitating factor for the development of postoperative ED in children.''
- “It was determined that the longer the solid food fasting time, the longer the postoperative ED period.” - from the statistical methodology you have used and presented, this conclusion cannot be drawn.
----- The statistical methodology we used has been corrected and added to the statistical analysis section.
è The same here. See above.
----- ¥ ¥ This statement ‘’It was determined that the longer the solid food fasting time, the longer the postoperative ED period.” is removed from manuscript
- From my point of view you should address the generalizability of your results in the discussion. While you have worded it very nicely: “In our study, pediatric patients who underwent MRI under sedation were selected in order to exclude conditions affecting the development of postoperative ED, such as general anesthesia, major surgery, large area resection, prolonged anesthesia time, presence of postoperative pain, and opioid use to examine the relationship between preoperative fasting time, last feeding time with solid and liquid food, FBG and postoperative ED.” - your argumentation is not very convincing. You should discuss to what extent your results from a cohort with an intervention that lasted only about 19 minutes and received only sedation is transferable to real-life situations in which ED is highly relevant. In the discussion, please also address in what way the patients' underlying diseases influenced the results. Patients who require MRI examinations are a very selective patient group. This should be taken into account at least in a simple group comparison and, if necessary, also in the regression analysis.
----- The statistical methodology we used has been corrected and added to the statistical analysis section.
è That's okay from my point of view.
- Please follow the general guidelines for presenting the results of a cohort study. Please, apply to the STROBE checklist (include a FlowChart, state the time period of the study conduction, description of all variables in the method section) (corrected)
---- ---- Necessary arrangments have been made.
è Unfortunately, a description of all collected items is still missing in the method section.
Minors:
Abstract:
- Please include a description of the statistical methods in the abstract.
è This did not happen. Nor was the abstract as a whole revised after the statistics were adjusted. This is not satisfactory.
---- The material method section of the abstract was edited as follows;
110 pediatric patients were included in the study. Preoperative fasting (solid-fluid) time and FBG were recorded. The development of ED in the patients who underwent MRI under sedation was evaluated with the Paediatric Anaesthesia Emergence Delirium (PAED) value for 30 minutes every 5 minutes in the recovery room. PAED score of ≥10 was grouped as having ED, PAED score of <10 without ED at any time. The PAED scores compared with other variables, ASA, age, weight, MRI examination time, and FBG level and fasting times. The risk factors affecting the occurrence of ED were examined.
- Please state units of fasting times.---- stated.
- Please correct the mean age. This must be a typo.--- Corrected.
Methods:
- please specify more in detail “An anesthesiologist […] giving information about preoperative fasting times”.. ----- fasting times were given clearly.
Results:
- Please state incidence of postoperative ED in the results section. ---- Added to the result section.
- Please correct the mean age. This must be a typo.---- Corrected.
- According to Table 1: how can it be that there were patients with an ASA PS of 3 when patients with ASA PS of 3 and 4 were excluded?---- corrected
- According to Table 1: Please replace gender with sex (I assume you have examined biological sex). ---- corrected (corrected)
è Unfortunately, you continue to use "gender" in the manuscript
- Please provide more content on the “Drug Use” and “Comorbidities variables”. As it is presented now, it is not possible to get an impression of the cohort. ---- excluded from the table because it did not contribute to the cohort
Further Minors:
- Please also provide number and percentages of ED (+) and ED (-) in Table 3. (corrected)
- Please do not use the * to annotate Mann Whitney U test. The * is very often equated to the significance level in publications. (corrected)
- You stated “In the literature, there are studies of preoperative fasting and postoperative ED in pediatric patients undergoing MRI, but the effect of the two factors on each other has not been evaluated.” Please provide adequate references.
---- Statement is corrected as;
In the literature, there are studies of preoperative fasting and postoperative ED in pediatric patients undergoing undergoing various surgeries, but the effect of the two factors on each other has not been evaluated (7,16).
- Please stay concise with the term “ED”. In some places you use the term “postoperative delirium (corrected)